# Old but Fancy: Curcumin in Ulcerative Colitis—Current Overview

**DOI:** 10.3390/nu14245249

**Published:** 2022-12-09

**Authors:** Aleksandra Pituch-Zdanowska, Łukasz Dembiński, Aleksandra Banaszkiewicz

**Affiliations:** Department of Pediatric Gastroenterology and Nutrition, Medical University of Warsaw, 02-091 Warsaw, Poland

**Keywords:** anti-inflammatory, inflammatory bowel disease, intestinal microbiota, polyphenols

## Abstract

Ulcerative colitis (UC) is one of the inflammatory bowel diseases (IBD). It is a chronic autoimmune inflammation of unclear etiology affecting the colon and rectum, characterized by unpredictable exacerbation and remission phases. Conventional treatment options for UC include mesalamine, glucocorticoids, immunosuppressants, and biologics. The management of UC is challenging, and other therapeutic options are constantly being sought. In recent years more attention is being paid to curcumin, a main active polyphenol found in the turmeric root, which has numerous beneficial effects in the human body, including anti-inflammatory, anticarcinogenic, and antioxidative properties targeting several cellular pathways and making an impact on intestinal microbiota. This review will summarize the current knowledge on the role of curcumin in the UC therapy.

## 1. Introduction

Ulcerative colitis (UC) is a chronic inflammatory bowel disease (IBD) that affects the colon [1]. Typical mucosal inflammation involves the rectum but could continuously extend to proximal segments of the large intestine. Like other types of IBD, UC is classified as an autoimmune disease of unclear etiopathology characterized by phases of exacerbation and remission. Its symptoms consist mainly of abdominal pain and bloody diarrhea. The etiology and pathogenesis of UC are multifactorial and still not fully understood; it includes genetic predisposition, immunological dysregulation, intestinal dysbiosis, epithelial barrier dysfunction, and many potential environmental factors, which jointly lead to sustaining chronic inflammation [2].

The incidence of UC is rising around the world causing a global problem, and it is being diagnosed at an earlier age. It is estimated that at least a quarter of patients experience their first symptoms in childhood or adolescence [3,4]. Moreover, extensive colitis occurs in two-thirds of newly-diagnosed pediatrics patients versus in only 20–30% of adult patients [5]. The population-based studies of Ng et al. show that in adults the highest prevalence of UC is in Europe and in North America (505/100,000 in Norway and 286/100,000 in the USA) [4]. In the pediatric population, UC prevalence is estimated at 22/100,000 in most European and North American regions [6].

The main goal of UC management is to induce and maintain remission, defined as the resolution of symptoms with endoscopically confirmed mucosal healing, as long as possible. Long-term maintenance of IBD remission enables children to grow and develop properly, and adults to lead a normal personal and professional life. Pharmacological treatment for UC depends on the disease extent and the degree of its clinical activity and includes 5-aminosalicylic acid drugs, glucocorticoids, and immunomodulating, immunosuppressive, and biological drugs. Unfortunately, monotherapy is not always fully effective and the long-term combination of several drugs may be associated with the occurrence of side effects. Surgical management, with colectomy as the most common procedure, is sometimes the only solution in relapsed and severe disease and is usually being implemented in patients with pancolitis [1,7]. There is no fully effective, universal treatment as the etiology of the disease is complex. However, the therapy should aim to provide satisfactory quality of life for the affected individuals.

These days, more attention is being paid to the necessity of modifying the environmental factors, which includes, apart from the others, dietary aspects, as the so-called Western diet has been linked to the highest prevalence of IBD, including UC [8]. Such a strategy may improve clinical outcomes in UC patients while minimizing the risk of the occurrence of side effects. Recently, alternative therapeutic options that are being explored include specific dietary approaches or usage of nutraceuticals (e.g., polyphenols). A nutraceutical is “a food (or part of a food) that provides medical or health benefits, including the prevention and/or treatment of a disease” [9,10].

More research is focusing on the medicinal value of polyphenols to prevent immune-mediated intestinal chronic inflammation. Over the last few years curcumin, a natural polyphenol belonging to the curcuminoid family (compounds derived from *Curcuma longa* L. [turmeric root]), is of greater interest in the context of managing UC. It seems that curcumin is a promising natural compound due to its widely described multi-beneficial effects on microbiota alteration and antioxidative, antitumor and—the most relevant—anti-inflammatory properties. The anti-inflammatory effect is mainly mediated via the regulation of the nuclear factor kappa-light-chain-enhancer of activated B cells (NF-κB) which results in the inhibition of proinflammatory cytokines, such as IL-1, IL-6, and TNF-α expression [11,12,13].

This article provides an overview and clinical perspectives of the role of curcumin usage as adjuvant therapy in ulcerative colitis, with particular attention to its influence on intestinal microbiota.

## 2. Curcumin

Curcumin, also known as the ‘golden spice of India’, has been used for thousands of years as an essential medicinal, herbal ingredient that exhibits anti-inflammatory, antioxidant, or antimicrobial properties, mainly. It is also well-known in Chinese traditional medicine. Nowadays, curcumin, an orange-yellow crystalline powder, is widely used in the food industry mostly as a dye (E100) in foodstuffs and beverages processing. It is also a very popular dietary spice in many cuisines worldwide. It is extracted from the rhizomes of *Curcuma longa* L. from the ginger family *Zingiberaceae*. Curcumin comprises 2–5% of the rhizome content. Chemically, curcumin is a diferuloylmethane or 1,7-bis (4-hydroxy-3methoxyphenyl)-1,6-heptadiene-3,5-dione, with the molecular formula C_21_H_20_O_6_ [14]. It is the principal curcuminoid, and the most active component of the total turmeric spice. It belongs to substances generally recognized as safe (GRAS), with its safety and tolerance confirmed in human clinical trials [14,15,16]. The Joint Food and Agriculture Organization (FAO)/World Health Organization (WHO) Expert Committee on Food Additives (JECFA) and the European Food Safety Authority (EFSA) allocated an acceptable daily intake (ADI) for curcumin of 0–3 mg/kg body weight [17]. Commercial curcumin consists of three major compounds, which are referred collectively as curcuminoids: curcumin [diferuloylmethane] (82%), demethoxycurcumin (15%, DMC), and bisdemethoxycurcumin (3%, BDMC) [18].

Curcumin is a small molecular weight compound that is lipophilic, thereby nearly insoluble in aqueous physiologic media but is soluble in methanol, dimethylsufoxide, ethanol, and acetone, as well slightly soluble in benzene and ether. It is a very photosensitive compound [14,19]. This yellow-colored polyphenol is a small hydrophobic molecule that can accumulate in cell membranes, which are hydrophobic regions, and perform as an antioxidant, scavenging reactive oxygen species. It is stable in the range of pH between 2.5 and 6.5 and it remains quite stable at the low acidic pH of the stomach [20,21].

After oral administration, curcumin is rapidly metabolized via reduction, sulfation, and glucuronidation in the liver, kidneys, and intestinal mucosa with low absorption of accumulated curcumin from the intestine [22,23]. Phase I of curcumin metabolism consists of reduction of its double bonds in hepatocytes and enterocytes, transforming it to dihydrocurcumin, tetrahydrocurcumin, hexahydrocurcumin, and octahydrocurcumin [24,25]. Phase II consists of conjugation of glucuronide or sulfate to the curcumin and to its hydrogenated metabolites in the intestinal and hepatic cytosol [26]. Major curcumin metabolites in plasma, curcumin glucuronide and sulphate conjugate metabolites, are characterized by low activity [22]. There is a greater curcumin metabolic conjugation and reduction in the human models than in rat models. Therefore, human clinical trials are much more appropriate, and are constantly, highly needed to assess the real curcumin therapeutic potential [26]. The gut microbiome is considered capable of deconjugating phase II metabolites and converting them back to the metabolites of phase I. This process can also lead to the production of, for example, ferulic acid (4-hydroxy-3-methoxy-cinnamic acid) which is a phenolic antioxidant compound that has a high radical scavenger effect for free radicals [27,28]. Furthermore, it was found that commensal *Escherichia coli* had the highest metabolizing activity among curcumin-converting microorganisms via an enzyme called “NADPH-dependent curcumin/dihydrocurcumin reductase” (CurA). *E. coli* acts in a two-step reduction process, converting curcumin to dihydrocurcumin, and then to tetrahydrocurcumin [29]. In another study it was reported that *Blautia* sp. *MRG-PMF1* carries out an alternative metabolism of curcumin which is curcumin demethylation. This process led to the production of two metabolites, which were demethylcurcumin and bisdemethylcurcumin [30].

For years, the main limitations of curcumin as a therapeutic option have been its chemical instability and poor systemic bioavailability, with very low or almost undetectable concentrations in blood and extraintestinal tissues, and its rapid metabolism and prompt systemic elimination [31]. Rapid elimination of curcumin from the body results in the excretion of more than 90% of curcumin in the feces. The search for an appropriate delivery method has been a challenge to curcumin use as an effective treatment for specific diseases. This has resulted in the development of specific, promising strategies with some success in increasing blood concentrations and a few examples are mentioned below [32].

The most common way to increase curcumin’s poor pharmacokinetic profile is the combination of curcumin with the natural alkaloid of black pepper—piperine (*Piper nigrum*) that is a strong inhibitor of glucuronidation process, mainly. This formulation resulted in 3-fold increase of curcumin concentrations, as compared to pure curcumin, when 5 mg of piperine was added to 2 g of curcumin [31]. Curcumin dispersed with colloidal nanoparticles (Theracurmin) was highly absorptive—the area under the blood concentration-time curve (AUC) when administered orally 30 mg of Theracurmin was 27-fold higher than that of curcumin powder in healthy volunteers [33]. Another example of curcumin’s bioavailability improvement is curcumin in a micellar system. In healthy subjects, the administration of micronized curcumin powder and curcumin incorporated into liquid micelles resulted in a significantly higher concentration of curcumin in plasma and in urine, than after supplementation of native curcumin [34]. There are also some other curcumin combinations like a mixture of turmeric powder and turmeric essential oil, lipid-curcumin formulations, or curcumin mixture with lecithin [24].

## 3. Curcumin, Anti-Inflammatory Effect, and Ulcerative Colitis

The significant anti-inflammatory properties of curcumin, being described over the years have attracted a lot of researchers’ interest, especially in the context of treating diseases with a chronic inflammation basis. NF-κB is a multi-functional key nuclear transcription factor, involved in the development of inflammatory diseases. It is believed to strongly affect the progression of mucosal inflammation in ulcerative colitis. In many studies it has been shown that curcumin inhibits NF-κB expression by blocking IkappaB (IκB) kinase, that leads to the prevention of cytokine-mediated phosphorylation and the degradation of IκB, which is an NF-κB inhibitor. Hence, the expression of proinflammatory cytokines, such as IL-1, IL-6, IL-8, and TNF-α, is inhibited [35,36]. Furthermore, it was also reported that curcumin inhibited the activity of proinflammatory proteins (e.g., activated protein-1, peroxisome proliferator-activated receptor gamma, transcription activators, the expression of β -catenin) [37].

## 4. Curcumin, Intestinal Microbiota, and Ulcerative Colitis

As oral supplementation with curcumin leads to its high concentration in the gastrointestinal tract, studies have slowly focused on its impact on the intestinal microbiota. Via this mechanism, the problem of low systemic curcumin bioavailability probably is not a significant issue within the gastrointestinal tract, and curcumin may have a hypothetical beneficial influence on the gut microbiome [21,38]. A bidirectional interaction exists between curcumin and gut microbiota. Gut microbiota are actively involved in curcumin metabolism, which lead to curcumin biotransformation (demethylation, hydroxylation, demethoxylation) and the production of metabolites. Curcumin supplementation is effective in promoting the growth of beneficial bacterial strains, improving intestinal barrier functions, and counteracting the expression of pro-inflammatory mediators [39].

Only one study in healthy humans assessed microbiota alteration after oral curcumin administration. Peterson et al., in a double-blind, randomized, placebo-controlled pilot study with 30 healthy subjects, assessed changes in the gut microbiota using 16S rDNA sequencing after oral supplementation with turmeric 6000 mg with extract of piperine, curcumin 6000 mg with Bioperine (black pepper extract) tablets, or placebo, at baseline and after 4 and 8 weeks. They found that both turmeric and curcumin in a highly similar manner altered the gut microbiota. Participants who took turmeric supplementation displayed a 7% increase in observed microbial species post-treatment, and curcumin-treated subjects displayed an average increase of 69% in detected bacterial species. The authors indicated that the intestinal microbiota responses to such therapy were highly personalized. Subjects defined as “responders” showed uniform increases in most *Clostridium* spp., *Bacteroides* spp., *Citrobacter* spp., *Cronobacter* spp., *Enterobacter* spp., *Enterococcus* spp., *Klebsiella* spp., *Parabacteroides* spp., and *Pseudomonas* spp., with reduction in the relative abundance of several *Blautia* spp. and most *Ruminococcus* spp. [40].

UC patients have been reported to have intestinal dysbiosis with regard to the diversity and composition of intestinal microbiota on different taxonomic levels. Bacteria observed to decrease belong mainly to *Firmicutes* and *Bacteroidetes* phyla, which are considered beneficial, and increased bacteria belong to the *Enterobacteriaceae* family, considered harmful [41,42,43]. Recently, in a pilot study, Zakerska-Banaszak et al. determined specific changes in the gut microbiota profile in Polish UC patients compared to the healthy control group. They reported significantly lower gut microbiome diversity in UC patients with more abundance of *Proteobacteria* (8.42%), *Actinobacteria* (6.89%), and *Candidate Division TM7* (2.88%) compared to the controls. They also found that *Bacteroidetes* and *Verrucomicrobia* were less abundant in UC as compared to the control group (14% and 0% vs. 27.97% and 4.47%, respectively). Additionally important, the researchers observed a decrease in beneficial bacteria, such as *Faecalibacterium prausnitzii* and *Blautia*, the butyric acid producers. Butyric acid is one of the crucial short-chain fatty acid (SCFA) for maintenance of intestinal homeostasis and is produced from specific dietary fibers [44]. It is uncertain whether intestinal dysbiosis is a consequence or a cause of chronic gut inflammation. However, studies have pointed out the pivotal role of gut microbiota in the host immune system via gut-associated lymphoid tissue (GALT). Therefore, any microbiome disturbances may be related to many diseases, including ulcerative colitis, and studies are focusing on the effects of intestinal dysbiosis on colonic mucosal barrier integrity, regulation of the host immune system response, and finally, on contributing to the progression of tumorigenesis [45]. Gut microbiota modifications through dietary interventions, targeting bacteria species involved in the disease course, may broaden the therapeutic landscape, providing a chance for personalized therapies in UC patients [46].

According to our knowledge, to date, no human studies have been published to assess the effect of curcumin on the gut microbiome in patients with UC. We reported two studies on the effect of curcumin supplementation on the intestinal microbiota in experimental animal models of colitis.

The first study assessed the effect of a curcumin-supplemented diet (98.05% pure curcumin, free of contaminating curcuminoids: demethoxycurcumin and bisdemethoxy-curcumin) in a mouse model of colitis-associated colorectal cancer, which resulted in increased bacterial richness, prevented the age-related decrease in alpha diversity, increased the relative abundance of *Lactobacillales*, and decreased *Coriobacteriales*. The authors concluded that the favorable effect of curcumin on tumorigenesis was linked with the maintenance of a greater colonic microbiota diversity [47].

The second study, conducted by Ohno et al., examined the effects of nanoparticle curcumin (Theracurmin) supplementation on dextran sulfate sodium (DSS)-induced colitis in mice. The curcumin therapy increased the abundance of butyrate-producing bacteria, which led to increased fecal butyrate levels [48]. Butyric acid is known to have a beneficial influence on intestinal homeostasis and energy metabolism. Its anti-inflammatory properties have been linked with enhancing intestinal barrier function and mucosal immunity [49].

## 5. Curcumin for Induction or Maintenance of Remission in UC: Data from Systematic Reviews and Meta-Analysis

Since 2020, the interest in curcumin for treating UC has increased noticeably, as evidenced by the increase in published systematic reviews. Searching via PubMed the descriptors “curcumin and ulcerative colitis” yielded sixteen systematic reviews and/or meta-analyses since 2012, where nine of them have been published from 2020 to the present. Four of these nine papers were not summarized in our review: a protocol from Yang et al. (2020) for a planned systematic review and meta-analysis, an assessment of curcumin supplementation in various gastrointestinal diseases from Atefi et al. (2021), a systematic review from Goulart et al. (2021) which also included studies with patients with Crohn’s disease, and a systematic review and meta-analysis that included patients with another autoimmune disease [50,51,52,53].

Chandan et al. reviewed and analyzed seven clinical trials which included 380 UC patients (curcumin *n* = 188; placebo *n* = 190) ranging in age from 32.7 ± 8.9 years to 45.2 ± 15.8 years [54]. The pooled odds ratio (OR) for clinical remission with curcumin use was 2.9 (95% CI 1.5–5.5, *I*^2^ = 45, *p* = 0.002), for clinical response 2.6 (95% CI 1.5–4.5, *I*^2^ = 74%, *p* = 0.001), and for endoscopic response/remission 2.3 (95% CI 1.2–4.6, *I*^2^ = 35.5%, *p* = 0.01). The authors concluded that combined therapy based on mesalamine plus curcumin was linked with almost 3-fold better clinical response odds than the placebo group.

Zheng et al. reviewed six RCTs with a total of 349 UC patients (curcumin *n* = 173, control group *n* = 176) that showed that curcumin plus mesalamine was effective in inducing clinical remission (OR = 5.18, 95% CI 1.84–14.56, *p* = 0.002), endoscopic remission (OR = 5.69, 95% CI: 1.28–25.27, *p* = 0.02), and endoscopic improvement (OR = 17.05, 95% CI 1.30–233.00, *p* = 0.03), but not clinical improvement (OR = 4.79, 95% CI: 1.02–22.43, *p* = 0.05) [55]. The authors concluded that curcumin with mesalamine in UC patients was effective and safe. They recommended further studies to determine appropriate drug form, dose, duration, and delivery method.

Coelho et al., in their systematic review, included six RCTs with a total of 372 patients (aged 23 to 61) with UC [56]. Curcumin was administered for the maintenance or induction of remission in patients with mild to moderate disease activity. The studies showed good tolerance to curcumin complementary therapy administered with standard treatments. In addition, five of six trials demonstrated good results related to the clinical and/or endoscopic remission/response.

Goulart et al. assessed the role of curcumin therapy for the induction of remission in UC in their meta-analysis, which included only four RCTs with 238 enrolled patients with mild-to-moderate UC [57]. The authors concluded that despite the small number of patients enrolled, the supplementation of curcumin had a beneficial impact on the clinical remission of patients with UC.

The latest systematic review about the efficacy and safety of supplemental curcumin therapy in UC, performed by Yin et al. (2022), included six randomized trials with total number of patients *n* = 385 [58]. The authors reported that supplemental curcumin treatment for UC was safe without any severe side effects. It effectively induced clinical remission (RR = 2.10, 95% CI 1.13 to 3.89) but not clinical improvement, endoscopic remission, or endoscopic improvement. The authors emphasized the importance of the optimal method of curcumin administration for a better curative effect and suggested further well-planned studies. In Table 1 we summarize the basic characteristics of the above-described five studies.

Shi et al. critically assessed the scientific quality of seven relevant systematic reviews (SRs) and meta-analyses (Mas), including those by Chandan, Zheng, and Goulart (above) [59]. They found the methodological quality of all assessed SRs/Mas to be very low. The quality of evidence for outcomes ranged from moderate to very low. Factors in the low-quality assessments included imprecision, publication bias, and inconsistency. The authors of this overview of SRs state that curcumin may be effective and safe for treatment of UC, but that further research is needed.

## 6. Curcumin in the Algorithm of Managing Pediatric UC

According to official recommendations from the European Crohn’s and Colitis Organization (ECCO) and the European Society of Paediatric Gastroenterology, Hepatology and Nutrition (ESPGHAN), curcumin may be considered an additional therapy for inducing and maintaining clinical remission in mild to moderate (PUCAI ≤ 60) disease, assessed by the Paediatric Ulcerative Colitis Activity Index (PUCAI) [60]. Curcumin has been shown to be well tolerated in children. It has been established that the safe curcumin dose for remission induction is up to 4 g/day and for maintenance up to 2 g/day. Proposed dosage is shown in Table 2 [61,62,63].

## 7. Conclusions and Directions for Future Research

Data is accumulating on curcumin’s anti-inflammatory effect in patients with UC, but there is still insufficient evidence to define its effect on the composition and functioning of intestinal microbiota in the course of the disease. For some individuals affected by UC, there seems to be a real need to identify curcumin’s role as a supplement in safe, bioavailable, tolerated doses, and to incorporate it into routine clinical practice for better clinical outcomes and improvement of the quality of life of patients. Well-planned, large-scale, randomized controlled trials are needed to assess the benefits of such supplementation, including the impact on composition, diversity, and metabolic functions of intestinal microbiota, in a treatment scheme for the induction and/or in the maintenance of remission in pediatric and adult patients.

## Figures and Tables

**Table 1 nutrients-14-05249-t001:** Characteristics of the included systematic reviews and meta-analysis.

Authors (Year)	Number of Included Studies	Total Number of Subjects	Age of Subjects (Years)	Main Conclusions
Chandan et al. (2020) [54]	7	380	Mean age curcumin: 32.7 ± 8.9, placebo: 45.2 ± 15.8	Combined therapy based on mesalamine plus curcumin was linked with almost 3-fold better clinical response odds than the placebo group
Zheng et al. (2020) [55]	6	349	Range of age: 18–75	Curcumin, as an adjuvant treatment of mesalamine, was proved to be effective and safe in UC
Coelho et al. (2020) [56]	6	372	Range of age: 23–61	Curcumin was well tolerated and was not associated with any serious side effects. Curcumin could be a safe, effective therapy for maintaining remission in UC when administered with standard treatments
Goulart et al. (2020) [57]	4	238	Range of age: 18–70	Supplementation of curcumin had a beneficial impact on the clinical remission of patients with UC
Yin et al. (2022) [58]	6	385	N/A	Supplemental curcumin treatment for UC was effective for clinical remission without causing severe side effects. The appropriate methods of administration can achieve better curative effect.

**Table 2 nutrients-14-05249-t002:** Proposed dosage of curcumin in pediatric patients according to ECCO/ESPGHAN [60].

Induction of Remission (Daily, 2 Divided Doses)		Maintenance
children over 30 kg	4 g/d	Induction doses may be halved
children 20 to 30 kg	3 g/d	
under 20 kg ^1^	2 g/d	

^1^ Safety has not been established in infants.

## Data Availability

The data obtained during the preparation of the manuscript will be available from the first author on a justified request.

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
