# Peer review of "Old but Fancy: Curcumin in Ulcerative Colitis—Current Overview"

_nutrients, 2022, doi:10.3390/nu14245249_

Round 1

Reviewer 1 Report

In this review entitled “Old but Fancy: Curcumin in Ulcerative Colitis – Current Overview”, the authors described the protective role of curcumin in UC patients. Curcumin is an active polyphenol from the curcuminoid family derived from Curcuma longa L. (Turmeric root). It is well known in the nutraceutical field for its anti-inflammatory, anticarcinogenic, and antioxidative properties. This review focuses on the role of curcumin in UC therapy due to its anti-inflammatory role and positive impact on the intestinal microbiome.

In my opinion, this manuscript is quite complete and well-written, however, I would like to give a couple of suggestions to improve the work:

1.     The creation of a graphical abstract to best represent the anti-inflammatory effect of curcumin.

2.     To summarize the main findings of the analyzed literature in a table of contents: this could further facilitate the reading and make it more intuitive and enjoyable. 

Author Response

Dear Reviewer,

Thank You for your review. We are grateful for your opinion. 

According to your comment, we put some characteristics of the analysed literature in a table. We agree that graphical abstracts enhance the attractiveness of manuscripts. However, we lack the skills required to create professional graphics.

Unless we can count on the Editor's support in this matter?

Reviewer 2 Report

In this review article Pituch-Zdanowska et al. summarized the current knowledge on the role of curcumin in the ulcerative colitis therapy. This article provides a balanced view of the topic, it is well prepared, however the novelty is not very high. This topic has been undertaken in many others reviews. But in my opinion it summarize the latest findings on the topic thus it can be considered as a valuable work. Basically all used literature items are properly selected to the topic of the work and are up-to-date (from the past 2–5 years). Overall I had a great pleasure reading this manuscript, as it provides a novel point of view and is very carefully prepared.

Author Response

Dear Reviewer,

Thank you very much for your review. We are grateful for your appreciation.